# The Analytical Strategy of “Ion Induction and Deduction Based on Net-Hubs” for the Comprehensive Characterization of Naringenin Metabolites In Vivo and In Vitro Using a UHPLC-Q-Exactive Orbitrap Mass Spectrometer

**DOI:** 10.3390/molecules27217282

**Published:** 2022-10-26

**Authors:** Yi-Fang Cui, Wen-Wen Zhang, Ya-Nan Li, Jing Xu, Xian-Ming Lan, Shu-Yi Song, Yong-Qiang Lin, Long Dai, Jia-Yu Zhang

**Affiliations:** 1College of Pharmacy, Binzhou Medical University, Yantai 264003, China; 2College of Pharmacy, Shandong University of Traditional Chinese Medicine, Jinan 250300, China; 3Shandong Provincial Institute for Food and Drug Control, Jinan 250101, China

**Keywords:** naringenin metabolites, metabolic pathway, analytical strategy, UHPLC-Q-Exactive Orbitrap mass spectrometer

## Abstract

Naringenin (5,7,4′-trihydroxyflavanone), belonging to the flavanone subclass, is associated with beneficial effects such as anti-oxidation, anticancer, anti-inflammatory, and anti-diabetic effects. Drug metabolism plays an essential role in drug discovery and clinical safety. However, due to the interference of numerous endogenous substances in metabolic samples, the identification and efficient characterization of drug metabolites are difficult. Here, ultra-high-performance liquid chromatography (UHPLC) coupled with high-resolution mass spectrometry was used to obtain mass spectral information of plasma (processed by three methods), urine, feces, liver tissue, and liver microsome samples. Moreover, a novel analytical strategy named “ion induction and deduction” was proposed to systematically screen and identify naringenin metabolites in vivo and in vitro. The analysis strategy was accomplished by the establishment of multiple “net-hubs” and the induction and deduction of fragmentation behavior. Finally, 78 naringenin metabolites were detected and identified from samples of rat plasma, urine, feces, liver tissue, and liver microsomes, of which 67 were detected in vivo and 13 were detected in vitro. Naringenin primarily underwent glucuronidation, sulfation, oxidation, methylation, ring fission, and conversion into phenolic acid and their composite reactions. The current study provides significant help in extracting target information from complex samples and sets the foundation for other pharmacology and toxicology research.

## 1. Introduction

Flavonoids, consisting of two aromatic rings with six carbon atoms interconnected by a heterocycle with three carbon atoms (rings A, B, and C), are essential components in the human diet, even though they are not considered nutrients [1]. With the chemical name of 5,7,4′-trihydroxyflavanone, naringenin is widespread in some edible fruits and vegetables, such as *Citrus* species, tomatoes, and figs. More importantly, evidence from epidemiological, clinical, and preclinical studies has demonstrated that naringenin is responsible for various pharmacological activities such as anti-oxidation [2,3], anticancer [4,5,6], anti-inflammatory [7,8], and anti-diabetic [9] effects, cardiovascular protection [10,11], the regulation of immunity [12], the protection of the nervous system [13], and the prevention of diet-induced weight gain [14]. Naringenin may exert therapeutic effects against COVID-19 by inhibiting the main protease of COVID-19, 3CLpro, and by reducing the activity of ACE2 receptors [15].

The drug disposition in the body involves absorption, distribution, metabolism, and excretion (ADME). Drug metabolism is a complex biotransformation process in which drugs are structurally modified to different molecules by diverse enzyme families, such as CYP450s, dehydrogenases, and flavin-containing monooxygenases [16,17]. The primary purpose of metabolism is to clear endogenous and/or exogenous molecules from the body by converting lipophilic chemicals into hydrophilic products to facilitate elimination. The products of metabolism are known as metabolites, either pharmacologically active or inactive. Revealing the metabolic properties of drugs could contribute to deeper understanding of the effectiveness of drugs, changing the in vivo half-life and risk-benefit ratio of a drug, and playing a significant role in drug design. In general, metabolic reactions are classified into ‘phase I’ and ‘phase II’ metabolic reactions. As for the former, drugs expose or introduce groups such as -OH, -COOH, and -NH_2_ through oxidation, reduction, and hydrolysis reactions. During ‘phase II’ drug metabolism, which mainly includes sulfated, glucuronidated, and methylated biotransformation reactions, the drugs usually conjugate with a hydrophilic endogenous molecule to facilitate the elimination of drugs from the body smoothly. The liver is the primary organ responsible for drug metabolic processes, while liver microsomes, subcellular fractions derived from the endoplasmic reticulum of hepatic cells, are the dominant liver models in vitro and the default tools for drug discovery and development [18,19]. Some preliminary studies have been carried out on the metabolic process of naringenin and interrelated components. The pharmacokinetic parameters of naringin and naringenin were calculated by single-dose studies and multiple-dose studies, and 12 metabolites detected in liver microsomes were also analyzed [20]. Naringenin chalcone metabolites in rat plasma and urine were identified by LC–MS and NMR [21]. Moreover, naringenin-7-*O*-glucuronide and narigenin-4′-*O*-glucuronide, phase II metabolites of naringenin, were reported to perturb macrophage gene expression [22]. However, information obtained on the metabolic process of naringenin in the body is insufficient, suggesting that a comprehensive analysis of the transformation mechanism of naringenin is still needed.

The detection instrumentation is the critical factor in determining the results of metabolite identification. Liquid chromatography coupled with mass spectrometry has played a fundamental role as the predominant platform for metabolism studies ever since the introduction of atmospheric ionization techniques [23]. Subsequent information filtering and structural identification are the most significant procedures in the process of metabolite identification, playing a pivotal role in drug discovery and development [24]. However, partly due to the significant excess of endogenous material, detecting and characterizing drug metabolites in the complex biological matrices are difficult.

In this study, a UHPLC-Q-Exactive Orbitrap mass spectrometer, characterized by a high resolution, high-quality accuracy, wide quality range, and wide dynamic range, was adopted to collect naringenin metabolite data from rat plasma, urine, feces, liver tissue, and liver microsomes. It is worth mentioning that three methods (solid phase extraction (SPE), methanol precipitation, and acetonitrile precipitation) were used to process the plasma samples to obtain more comprehensive metabolite information. Moreover, a novel analytical strategy of “ion induction and deduction” based on the establishment of multiple “net-hubs”, an improvement of the previous analysis strategy [25], was proposed to improve the efficiency of metabolite identification. In the analytical strategy, the reaction pathways of metabolism were divided step by step due to the differences in naringenin biochemical modification in the metabolic process. Two major types of reactions (“reaction I“ and “reaction II“) were first proposed, and then a series of representative “net-hubs” were also established, which further refined the possible metabolic pathway and identified the attribution of fragment ions. Based on this analytical strategy, we distinguished a series of diagnostic product ion (DPI) groups, which were used to screen possible naringenin metabolites by rapidly matching the cleavage behavior of compounds in the samples. Ultimately, this facilitated the efficient capture of target metabolites from numerous endogenous metabolites. Furthermore, this strategy could also be deduced for metabolite identification of other similar compounds to increase work efficiency.

## 2. Results

In total, 78 metabolites (naringenin included) were detected and characterized, i.e., 31 in plasma, 43 in urine, 11 in feces, 8 in liver tissue, and 13 in liver microsomes. The detected metabolites are listed in Table 1.

### 2.1. The Establishment of an Analytical Strategy

Regarding the complexity of metabolites and the interference from endogenous material, establishing an accurate and efficient data analytical strategy to identify and characterize drug metabolites scientifically in complex organisms is of great importance. Therefore, an analytical strategy of “ion induction and deduction” based on metabolic “net-hubs” was proposed to systematically screen and identify naringenin metabolites in this work (Figure 1). In the analytical strategy, we comprehensively analyzed and mined the data in three steps. The induction of “DPI group I” of naringenin based on the mass fragmentation behaviors of naringenin in standard samples and metabolic samples was the first step. The second step was searching for the metabolites of naringenin based on the original nucleus of naringenin with the “DPI group I”, setting up related “net-hubs”, and inducing the “DPI group II”. When naringenin was metabolized by “reaction I” (phase II drug metabolism reactions and some phase I drug metabolism reactions such as a part of oxidation reaction), the mass spectral information of such metabolites matched well with that of naringenin. In contrast, the mass spectral information of metabolites produced by “reaction II” (other metabolic reactions, mainly including reduction, hydrolysis reactions, and so on) was not better matched with that of naringenin because their structures of naringenin were destroyed. Therefore, the third step was to construct “net-hubs” of such metabolic reactions and induce the “DPI group III” according to the occurrence characteristics of these reactions and related literature. Finally, all DPI groups were deduced, and various metabolites were screened by comparison with the DPI groups.

### 2.2. The Induction of “DPI Group I”

Naringenin belongs to the class of dihydro flavonoids, so Retro-Diels–Alder (RDA) cleavage and the neutral loss of H_2_O, CO, and CO_2_ occur easily in the process of MS fragmentation [26]. Other diagnostic product ions were generated due to the cleavage at different positions of the C-ring. In the ESI-MS/MS spectrum, naringenin gave rise to the [M-H]^−^ ion at *m*/*z* 271.06120 and generated the [M+H]^+^ ion at *m*/*z* 273.07575. To guide the subsequent rapid analysis, we induced and summarized six possible cleavage pathways of naringenin (Figure 2) by analyzing the cleavage behavior of the naringenin standard and combining relevant references [27,28]. In negative ion mode, through cleavage pathway A, product ions with high abundance at *m*/*z* 151 and *m*/*z* 119 were observed, which were associated with RDA cleavage. Similarly, according to the other cleavage pathways, product ions at *m*/*z* 145, *m*/*z* 125, *m*/*z* 177, *m*/*z* 93, *m*/*z* 165, *m*/*z* 107, and *m*/*z* 161 also appeared. Meanwhile, in positive ion mode, naringenin yielded a series of corresponding fragment ions such as *m*/*z* 153, *m*/*z* 121, *m*/*z* 179, *m*/*z* 96, *m*/*z* 147, *m*/*z* 163, and *m*/*z* 111 according to the cleavage pathways A, B, C, and E. The product ion at *m*/*z* 231 was also detected due to the cleavage pathway F in positive ion mode. In addition, the neutral loss of H_2_O (18 Da) and CO (28 Da) from parent ions also led to the generation of product ions at *m*/*z* 227 [M-H-CO_2_]^−^and *m*/*z* 255 [M+H-H_2_O]^+^. Finally, “DPI group I” was induced to be *m*/*z* 151, *m*/*z* 119, *m*/*z* 145, *m*/*z* 125, *m*/*z* 177, *m*/*z* 93, *m*/*z* 165, *m*/*z* 107, and *m*/*z* 161 in negative ion mode and *m*/*z* 153, *m*/*z* 121, *m*/*z* 179, *m*/*z* 96, *m*/*z* 147, *m*/*z* 163, and *m*/*z* 111 in positive ion mode.

### 2.3. The Induction of “DPI Group II” and the Establishment of Related “Net-Hubs”

Four main “net-hubs” were established according to the general laws of metabolic reactions. They were mono-oxidized, glucuronidated, sulfated, and methylated metabolites of naringenin. We found that these metabolites generated some other diagnostic production ions depending on the bound groups. For example, glucuronidated metabolites of naringenin might yield fragment ions at *m*/*z* 327 (*m*/*z* 151 + GluA), *m*/*z* 353 (*m*/*z* 177 + GluA), *m*/*z* 301 (*m*/*z* 125 + GluA), *m*/*z* 341 (*m*/*z* 165 + GluA), *m*/*z* 295 (*m*/*z* 119 + GluA), *m*/*z* 269 (*m*/*z* 93 + GluA), *m*/*z* 321 (*m*/*z* 145 + GluA), and *m*/*z* 337 (*m*/*z* 161 + GluA) in negative ion mode, or *m*/*z* 329 (*m*/*z* 153 + GluA), *m*/*z* 355 (*m*/*z* 179 + GluA), *m*/*z* 287 (*m*/*z* 111 + GluA), *m*/*z* 297 (*m*/*z* 121 + GluA), *m*/*z* 271 (*m*/*z* 95 + GluA), *m*/*z* 323 (*m*/*z* 147 + GluA), and *m*/*z* 339 (*m*/*z* 163 + GluA) in positive ion mode. Similarly, metabolites produced by other such metabolic reactions had similar fragmentation behavior. Therefore, “DPI group II” was induced to be *m*/*z* 151 + x (*m*/*z* 153 + x), *m*/*z* 177 + x (*m*/*z* 179 + x), *m*/*z* 125 + x (*m*/*z* 127 + x), *m*/*z* 119 + x (*m*/*z* 121 + x), *m*/*z* 145 + x (*m*/*z* 147 + x), *m*/*z* 93 + x (*m*/*z* 95 + x), *m*/*z* 161 + x (*m*/*z* 163 + x), and *m*/*z* 111 + x (x = molecular weight of substituent groups). If multiple binding reactions occurred, these fragment ions could be further converted to *m*/*z* 151 + nx (*m*/*z* 153 + nx), *m*/*z* 177 + nx (*m*/*z* 179 + nx), *m*/*z* 125 + nx (*m*/*z* 127 + nx), *m*/*z* 119 + nx (*m*/*z* 121 + nx), *m*/*z* 145 + nx (*m*/*z* 147 + nx), *m*/*z* 93 + nx (*m*/*z* 95 + nx), *m*/*z* 161 + nx (*m*/*z* 163 + nx), and *m*/*z* 111 + nx (x = the molecular weight of substituent groups, which can be different; n = 2,3,4 …). Combined with the Clog*P* value (compounds with a higher Clog*P* value show longer retention times in the C_18_ separation system), metabolic binding sites could be initially identified through these DPIs.

### 2.4. The Induction of “DPI Group III” and the Establishment of Other “Net-Hubs”

“Reaction II” may change the original nucleus structure of naringenin. By combining the literature, we found that apigenin and naringenin chalcone were important metabolites of naringenin and were set as the “net hub”. Based on the occurrence characteristics of these reactions and the relevant references, “DPI group III” was determined as *m*/*z* 151 ± x (*m*/*z* 153 ± x), *m*/*z* 177 ± x (*m*/*z* 179 ± x), *m*/*z* 125 ± x (*m*/*z* 127 ± x), *m*/*z* 119 ± x (*m*/*z* 121 ± x), *m*/*z* 145 ± x (*m*/*z* 147 ± x), *m*/*z* 93 ± x (*m*/*z* 95 ± x), *m*/*z* 161 ± x (*m*/*z* 163 ± x), and *m*/*z* 111 ± x (x = molecular mass difference due to the structural change of naringenin). These metabolites were further subject to other metabolic reactions; thus, their fragment ions could be further changed.

Furthermore, research has shown that 70% of ingested flavanones enter the colon where they are degraded by microbiota, principally producing small phenolic acid metabolites, and then they are absorbed into the circulatory system [29,30,31]. Thus, phenolic acid might be another meaningful metabolite of naringenin; thus, phenolic acid metabolites were also “net-hubs”. Further, they were identified individually by the existence of carboxyl groups, benzene rings, and so on. The other metabolites were directly classified in the metabolite part of naringenin due to their small amount, so no “net-hubs” could be established.

### 2.5. The Identification of Naringenin Metabolites

#### 2.5.1. The Identification of Naringenin Metabolites Produced by “Reaction I” (**M1**–**M40**)

Through the screening and matching the mass spectrometry information of potential metabolites in various biosamples using “DPI group I” and “DPI group II”, 40 metabolites of naringenin were unambiguously or tentatively identified.

**M1**, **M2**, **M3**, and **M4** showed their theoretical deprotonated molecular ions at *m*/*z* 624.13210 (C_27_H_28_O_17_, mass error within ± 5 ppm), which was 352 Da (2 GluA) higher than that of naringenin. Thus, they might be deduced as the diglucuronidated metabolites of naringenin. In their ESI-MS/MS spectra, fragment ions belonging to “DPI group II” at *m*/*z* 447 [M-H-GluA]^−^, *m*/*z* 449 [M+H-GluA]^+^, *m*/*z* 271 [M-H-2GluA]^−^, *m*/*z* 273 [M+H-2GluA]^+^, and *m*/*z* 175 [GluA-H]^−^ provided evidence for the identification of the glucuronic acid groups. Then, the product ions at *m*/*z* 151, *m*/*z* 119, and *m*/*z* 147 also further confirmed that their original nucleus was naringenin. Based on these findings, **M1**, **M2**, **M3**, and **M4** were preliminarily identified as the diglucuronidated metabolites of naringenin. In addition, the fragment ions at *m*/*z* 341 (*m*/*z* 165+ GluA) in negative ion mode and *m*/*z* 505 (*m*/*z* 153+ 2GluA) in positive ion mode were all observed in the ESI-MS/MS spectrum of **M4**, meaning that diglucuronidation was much more likely to occur on its A-ring. Taking **M4** as an example, its possible cleavage pathway is shown in Figure 3. Other metabolites were identified by similar methods, and their possible cleavage pathways are briefly described below.

**M5**, **M6**, **M7**, and **M8** generated a characteristic product ion at *m*/*z* 287 by the loss of GluA from the deprotonated [M-H]^−^ ion at *m*/*z* 463. Due to RDA cleavage, the fragment ions belonging to “DPI group I” at *m*/*z* 151 and *m*/*z* 119 were also observed, indicating the presence of the original nucleus of naringenin. Simultaneously, the fragment ion at *m*/*z* 287 suggested that they also underwent an oxidation reaction based on naringenin. Therefore, **M5**, **M6**, **M7**, and **M8** were all deduced as the oxidation and glucuronidated metabolites of naringenin. Furthermore, the glucosylated reaction might occur on their B-ring, while the oxidation reaction might occur on the A-ring according to the fragment ions belonging to “DPI group II” at *m*/*z* 337 (*m*/*z* 161+ GluA), *m*/*z* 193 (*m*/*z* 177+16 Da), *m*/*z* 167 (*m*/*z* 151+16 Da), and *m*/*z* 141 (*m*/*z* 125+16 Da).

**M9**, **M10**, and **M11** were 176 Da (GluA) larger than naringenin. In positive ion mode, the RDA fragmentation of the original nucleus produced fragment ions belonging to “DPI group I” such as *m*/*z* 153 and *m*/*z* 147. A train of fragment ions such as *m*/*z* 255 [M+H-GluA-H_2_O]^+^ and *m*/*z* 431 [M+H-H_2_O]^+^ also appeared in their ESI-MS/MS data. Correspondingly, **M10** also produced fragment ions similar to naringenin in negative ion mode. Therefore, **M9**, **M10**, and **M11** were all presumed to be glucuronidated products of naringenin.

The product ions belonging to “DPI group II” of **M12**, **M13**, and **M14** at *m*/*z* 271 [M-H-SO_3_]^−^, *m*/*z* 273 [M+H-SO_3_]^+^, *m*/*z* 207 [M-H-SO_3_-CO-2H_2_O]^−^, *m*/*z* 227 [M+H-SO_3_-CO-H_2_O]^+^, *m*/*z* 317 [M+H-2H_2_O]^+^, and *m*/*z* 335 [M+H-H_2_O]^+^ and fragment ions belonging to “DPI group I” at *m*/*z* 151, *m*/*z* 119, and *m*/*z* 177 demonstrated that **M12**, **M13**, and **M14** were sulfated metabolites of naringenin. Using ChemDraw to calculate their Clog*P* values, we speculated that the sulfated reactions of **M12**, **M13**, and **M14** occurred at C-5, C-4′, and C-7 position, respectively.

Five isomeric metabolites, **M15**, **M16**, **M17**, **M18,** and **M19**, were 16 Da larger than naringenin, producing fragment ions similar to those of naringenin. Hence, **M15**, **M16**, **M17**, **M18,** and **M19** were considered to be mono-oxidation metabolites of naringenin. Furthermore, the oxidation of **M16** and **M19** was presumed to occur on their A-rings due to the fragment ions belonging to “DPI group II” at *m*/*z* 167 (*m*/*z* 151+16 Da), *m*/*z* 193 (*m*/*z* 177+16 Da), and so on. Likewise, the oxidation of **M15**, **M17**, and **M18** was inferred to occur on their B-rings according to the fragment ions at *m*/*z* 109 (*m*/*z* 93+16 Da), *m*/*z* 135 (*m*/*z* 119+16 Da), and so on.

**M20** and **M21****,** 14 Da larger than naringenin, yielded product ions belonging to “DPI group II” at *m*/*z* 241 [M-H-CO_2_]^−^, *m*/*z* 253 [M-H-CH_3_OH]^−^, and *m*/*z* 205 [M-H-CO_2_-H_2_O]^−^ and fragment ions belonging to “DPI group I” at *m*/*z* 177 and *m*/*z* 151. Therefore, according to the above information, **M20** and **M21** were both considered methylation metabolites of naringenin.

**M22**, **M23**, and **M24** were 162 Da larger than the glucuronidated metabolites of naringenin. In their ESI-MS/MS spectra, the fragment ions belonging to “DPI group I” were observed. Then, a battery of fragment ions at *m*/*z* 433 [M-H-CO_2_-CO-H_2_O]^−^, *m*/*z* 591 [M-H-H_2_O]^−^, *m*/*z* 573 [M-H-2H_2_O]^−^, *m*/*z* 555 [M-H-3H_2_O]^−^, *m*/*z* 417 [M+H-GluA-H_2_O]^+^, and *m*/*z* 399 [M+H-GluA-2H_2_O]^+^ was generated by a series of neutral losses of H_2_O, CO_2_, CO, and GluA. All of the above-mentioned fragment ions indicated that **M22**, **M23**, and **M24** were glucuronidated and glucosylated metabolites of naringenin. Furthermore, the discovery of product ions belonging to “DPI group II” at *m*/*z* 313 (*m*/*z* 151+Glc) and *m*/*z* 489 (*m*/*z* 151+Glc+ GluA) in negative ion mode and at *m*/*z* 315 (*m*/*z* 151+Glc) in positive ion mode suggested that the transformation reaction took place on the A-ring. Similarly, according to the relative molecular mass change, the fragment cleavage behavior, and the identification process, **M12**, **M13**, and **M14** and the other five isomeric metabolites, **M25**, **M26**, **M27**, **M28**, and **M29**, were deduced to be sulfated and glucuronidated metabolites of naringenin. In addition, **M30** and **M31** were identified as the hydroxylated sulfated and glucuronidated metabolites of naringenin in the same way.

The fragment ions belonging to “DPI group I” of **M32** and **M33** at *m*/*z* 151, *m*/*z* 177, and *m*/*z* 271 in negative ion mode and at *m*/*z* 153, *m*/*z* 121, and *m*/*z* 179 in positive ion mode proved the existence of the nucleus of naringenin. In negative ion mode, the product ion at *m*/*z* 301 was attributed to the neutral loss of GluA. According to the difference in the relative molecular mass and molecular formula between **M32**, **M33**, and naringenin, **M32** and **M33** were presumed to be the hydroxylated glucuronidated and methylated metabolites of naringenin. The fragment ion at *m*/*z* 303 generated in positive ion mode also corroborated the inference. Therefore, **M34** and **M35** were believed to be the glucuronidated and methylated metabolites of naringenin. However, the distinction was that the glucuronidation in **M34** and **M35** might occur on their B-rings based on the fragment ions belonging to “DPI group II” at *m*/*z* 337 (*m*/*z* 161 + GluA), while the metabolic reaction of **M34** and **M35** could not be determined due to the limited information available.

**M36**,**M37**, **M38**, **M39**, and **M40** all underwent sulfated reaction on the original nucleus of naringenin, and thus the fragment ions of [M-H-SO_3_]^−^ or [M+H-SO_3_]^+^ were detected in these metabolites. In their ESI-MS/MS spectra, the fragment ions belonging to “DPI group II” at *m*/*z* 433 [M-H-SO_3_]^−^ and *m*/*z* 271 [M-H-SO_3_-Glc]^−^ were clearly detected. Hence, combined with the fragment ions belonging to “DPI group I” and ions at *m*/*z* 227 [M-H-SO_3_-Glc-CO_2_]^−^, **M36**, **M37**, and **M38** were interpreted as the sulfated and glucosylated metabolites of naringenin. Following the above reasoning process, **M39** and **M40** were characterized as the disulfated metabolites of naringenin.

#### 2.5.2. The Identification of Naringenin Metabolites Produced by “Reaction II” (**N1**–**N24**)

Through the screening and matching of the mass spectrometry information of samples using fragment ions belonging to “DPI group III”, 24 naringenin metabolites were determined.

**N2**, **N3**, and **N4** had the same theoretical [M-H]^−^ ions at *m*/*z* 269.04550 and [M+H]^+^ ions at *m*/*z* 271.06005, which were 2 Da smaller than naringenin. The product ions at *m*/*z* 153, *m*/*z* 147, and *m*/*z* 119 in positive ion mode and at *m*/*z* 151, *m*/*z* 107, *m*/*z* 93, and *m*/*z* 117 in negative ion mode were observed. Thus, **N2**, **N3**, and **N4** were deduced as hydroxylated and dehydrated products. Moreover, **N4** might be further deduced as apigenin [27]. The possible cleavage pathway of **N4** is shown in Figure 4. The possible cleavage pathways of other metabolites are also briefly described below.

The possible molecular formula of **N1** was C_15_H_1__3_O_5_ with mass errors of −4.828 ppm. The product ions belonging to “DPI group III” at *m*/*z* 179 [M-H-C_6_H_5_O]^−^ and *m*/*z* 125 [M-H-C_9_H_7_O_2_]^−^ were caused by the breakage of the carbon chain between the A-ring and B-ring. At the same time, the fragment ion at *m*/*z* 227 due to the neutral loss of CO and H_2_O was also detected. Thus, **N1** was speculated to be phloretin combined with its fragmentation behavior and related literature [27]. **N5** was 14 Da smaller than naringenin. In positive ion mode, the product ions belonging to “DPI group III” at *m*/*z* 149 [M+H-C_6_H_6_O_2_]^+^, *m*/*z* 121 [M+H-C_7_H_6_O_3_]^+^, and *m*/*z* 165 [M+H-C_6_H_6_O]^+^ could be attributed to the breakage of the carbon chain between the A-ring and B-ring. Thus, **N5** was considered to be *O*-desmethylangolensin or its isomers, one kind of ring-opening naringenin metabolites.

**N6** was preliminarily considered as the diglucuronidated metabolite of apigenin. Fragment ions belonging to “DPI group III” such as *m*/*z* 447 [M+H-GluA]^+^, *m*/*z* 271 [M+H-2GluA]^+^, and *m*/*z* 429 [M+H-GluA-H_2_O]^+^ could provide evidence for the mentioned inference. Likewise, **N15**, **N16**, and **N17** were identified as the glucuronidated metabolites of apigenin because of the occurrence of the fragment ions at *m*/*z* 269 [M-H-GluA]^−^, and *m*/*z* 271 [M+H-GluA]^+^. Simultaneously, the glucuronidated reaction of **N15** and **N16** might occur on their A-rings according to the fragment ions belonging to “DPI group III” at *m*/*z* 327 (*m*/*z* 151+176 Da) and *m*/*z* 311 (*m*/*z* 135+176 Da) in negative ion mode or *m*/*z* 329 (*m*/*z* 153+176 Da) and *m*/*z* 313 (*m*/*z* 137+176 Da) in positive ion mode. In addition, according to their Clog*P* values, it could be inferred that the glucuronidated reaction of **N15** occurred at the C5 position. Thus, the glucuronidated reaction of **N16** occurred at the C7 position, while that of **N17** was occurred at the C4′ position.

**N7** generated fragment ions at *m*/*z* 124, *m*/*z* 149, and *m*/*z* 93 by the breakage of the carbon chain between the A-ring and B-ring and was 26 Da smaller than apigenin. Meanwhile, it also yielded ions at *m*/*z* 225 by the neutral loss of H_2_O (18 Da). Therefore, **N7** was tentatively characterized as the decarbonylation metabolite of apigenin.

**N8**, **N9**, **N10**, and **N11** were 162 Da larger than apigenin, possessing a similar fragmentation behavior and thus were all preliminarily deduced to be glucosylated metabolites of apigenin, which may be genistin or its isomers. Fragment ions at *m*/*z* 163 and *m*/*z* 95 further confirmed the possibility.

The fragment ions of **N12** belonging to “DPI group III” at *m*/*z* 285 [M-H-GluA]^−^ and *m*/*z* 241[M-H-GluA-CO_2_]^−^ in negative ion mode and at *m*/*z* 287 [M+H-GluA]^+^ and *m*/*z* 269 [M+H-GluA-O]^+^ in positive ion mode were observed in the ESI-MS/MS spectra, indicating that **N12** might be the hydroxylated and glucuronidated metabolite of apigenin. Consistent with the above identification method, **N13** and **N14** were extrapolated as the hydroxylated and methylated metabolites of apigenin.

With the retention time of 11.42 min, **N18**, 14 Da larger than apigenin, yielded fragment ions belonging to “DPI group III” at *m*/*z* 133 [M+H-C_7_H_4_O_4_]^+^ and *m*/*z* 177 [M+H-C_7_H_8_O]^+^ in the ESI-MS/MS spectrum. Thus, **N18** was presumed to be the methylated metabolite of apigenin, and the transformation reaction might occur at the C4′ position.

**N19** and **N20** produced the theoretical [M-H]^−^ ion at *m*/*z* 285.04100 and the theoretical [M+H]^+^ ion at *m*/*z* 287.05555, 14 Da larger than apigenin. Therefore, **N19** and **N20** were deduced as mono-oxidation metabolites of apigenin due to product ions belonging to “DPI group III” at *m*/*z* 151 and *m*/*z* 107 in negative ion mode and at *m*/*z* 153 in positive ion mode.

**N21** was 176 Da larger than O-desmethylangolensin. In the ESI-MS/MS spectrum, the fragment ions at *m*/*z* 259 [M-H-GluA]^−^ provided evidence for identifying a glucuronide group and suggested the occurrence of the ring-opening reaction. The appearance of fragment ions belonging to “DPI group III” at *m*/*z* 149, *m*/*z* 121, and *m*/*z* 95 was due to the cleavage between the A-ring and B-ring. Moreover, ions at *m*/*z* 241 [M+H-H_2_O]^+^ and *m*/*z* 213 [M+H-CO-H_2_O]^+^ were also detected. Thus, **N21** was deduced as a glucuronidated metabolite of *O*-desmethylangolensin.

**N22** gave rise to the [M-H]^−^ ion at *m*/*z* 271.06146, and **N24** and **N25** gave rise to the [M+H]^+^ ion at *m*/*z* 273.07605 and *m*/*z* 273.07596 (mass error within ±5 ppm), meaning that they might be isomers of naringenin. In their ESI-MS/MS spectra, the product ions at *m*/*z* 227, *m*/*z* 151, and *m*/*z* 119 in negative ion mode and at *m*/*z* 153, *m*/*z* 147, and *m*/*z* 179 in positive ion mode supported our initial conjecture. However, the information obtained from mass spectrometry was limited. Thus, **N22**, **N23**, and **N24** were presumed to be isomers of naringenin, which may be present in naringenin chalcone [21].

#### 2.5.3. Identification of Phenolic Acid Metabolites

Combined with relevant literature [29,30] and fragment ions produced by carboxyl groups and benzene rings, 13 phenolic acid metabolites were finally identified.

In the ESI-MS/MS spectra of **H1**, **H2**, **H3**, and **H4**, the fragment ions at *m*/*z* 137 and *m*/*z* 121 were generated on account of the neutral loss CO and CO_2_, suggesting the presence of a carboxyl group. The fragment ions at *m*/*z* 147 [M-H-H_2_O]^−^ and *m*/*z* 103 [M-H-CO-H_2_O]^−^ confirmed the existence of hydroxyl substitution in their structures. Thus, **H1**, **H2**, **H3**, and **H4** could be deduced as 3-(4-Hydroxyphenyl) propionic acid or its isomers. Likewise, it was possible to presume the structures of **H11** and **H12**, which were eventually identified as *p*-hydroxybenzoic acid or its isomers.

Five isomeric metabolites, **H6**, **H7**, **H8**, **H9**, and **H10** were 2 Da less smaller than **H1**, **H2**, **H3**, and **H4**. Thus, they were preliminarily deduced to be the dehydrogenated products of **H1**, **H2**, **H3**, and **H4**. In their ESI-MS/MS spectra, the fragment ions at *m*/*z* 119 [M-H-CO_2_]^−^, *m*/*z* 121 [M+H-CO_2_]^+^, *m*/*z* 135 [M-H-CO]^−^, *m*/*z* 137 [M+H-CO]^+^, and *m*/*z* 119 [M+H-CO-H_2_O]^+^ provided evidence of the carboxyl group and hydroxyl group. Finally, **H6**, **H7**, **H8**, **H9**, and **H10** were all tentatively judged as *p*-hydroxycinnamic acid or its isomers.

**H13** with the molecular formula of C_9_H_10_O_2_ (mass error −1.506 ppm) was 16 Da smaller than **H1**, **H2**, **H3**, and **H4**. The fragment ions at *m*/*z* 121 [M-H-CO]^−^ and 105 [M-H-CO_2_]^−^ indicated that **H13** might be 3-phenylpropionic acid or its isomers.

Similar to the identification process of **H11** and **H12**, the fragment ions of **H5** at *m*/*z* 134 and *m*/*z* 160 in negative ion mode and at *m*/*z* 136 and *m*/*z* 162 in positive ion mode suggested the presence of a hydroxyl group and carboxyl group. Therefore, according to the accurate measurement, chromatographic behavior, cleavage fragmentation, and characteristic fragment ions, **H5** was characterized as hippuric acid or its isomer.

### 2.6. Possible Biotransformation Pathways of Naringenin

A total of 78 metabolites were finally identified in the study, 41 were detected by “reaction I” (naringenin included), and 37 were detected by “reaction II”, illustrated in Figure 5. The specific information of these metabolites is illustrated in Table 1. In the process of naringenin metabolism, conjugation reactions could occur on naringenin to produce its conjugation products such as glucuronidated metabolites, sulfated metabolites, methylated metabolites, and glucosylated metabolites. Naringenin could also be converted to naringenin chalcone, phloretin, *O*-desmethylangolensin, and multiple phenolic acid metabolites such as *p*-hydroxybenzoic acid, hippuric acid, and *p*-hydroxycinnamic acid by ring fission. In addition, naringenin was found to produce apigenin and other redox metabolites during the metabolic process. Then, naringenin formed multiple metabolic reaction chains through further multi-level metabolic reactions, and eventually, a complex biotransformation network was formed.

## 3. Discussion

### 3.1. Naringenin Metabolites In Vivo and In Vitro

In the present study, naringenin metabolites in vivo and in vitro were thoroughly investigated using a UHPLC-Q-Exactive mass spectrometer combining a novel analytical strategy of “ion induction and deduction” and multiple sample preparation methods (SPE, methanol precipitation, and acetonitrile precipitation). In total, 78 metabolites were finally identified; 67 metabolites were detected in vivo, and 13 metabolites were detected in vitro. Their distribution in each sample is illustrated in Figure 6A,B.

At the overall metabolite level, the main biotransformation pathways observed in vivo and in vitro were glucuronidation, sulfation, oxidation, methylation, ring fission, conversion into phenolic acid, and their secondary metabolic metabolism, which promoted the changes of the structure, variation of polarity, and biological properties of naringenin.

In the study of in vivo metabolites, we collected biological samples of plasma, urine, feces, and liver tissue from rats. In Figure 6A,B, it can be seen that most naringenin metabolites could be excreted by urine, implying that a urine sample might be a powerful sample for metabolite identification of naringenin. As shown in Figure 6A, the glucuronidated metabolites of naringenin, sulfated metabolites of naringenin, and phenolic acid metabolites were the major metabolites in the plasma and urine samples, while in feces, phenolic acids were mainly detected, meaning that phenolic acids might be the final metabolites of naringenin. It is worth mentioning that although the liver was the most important organ for metabolism in vivo, our study did not obtain large amount of information of metabolites from the liver tissue, and the metabolites obtained were all phenolic acids. It was preliminarily speculated that naringenin was metabolized rapidly in vivo (within 24 h), so other metabolites were not detected in the liver tissue after 24 h of administration.

Liver microsomes were used to conduct metabolic studies in vitro. In liver microsomes, we identified only 14 metabolites. However, apigenin (**M18**), an important metabolite of naringenin, was identified in liver microsome samples but not in plasma, urine, feces, or liver tissue samples, which also suggested that apigenin quickly underwent other metabolic reactions and then participated in the bodily physiological functions after being transformed.

The prototype of naringenin was only detected in plasma and liver microsomes, meaning that naringenin would not be excreted in prototype form. Apigenin, an edible plant-derived flavonoid, was proved to be one of the important metabolites of naringenin in this study. Certainly, apigenin is widely known for its anti-oxidant, anticancer, anti-inflammatory, anti-apoptotic, and anti-hyperglycemic effects [32,33], similar to the pharmacological activity of naringenin. However, apigenin also has biological properties that distinguish it from naringenin. For example, naringenin exerts its effect through a post-transcriptional mechanism to inhibit cytokine production, while apigenin mainly regulates cytokine production at the transcriptional level [8]. Furthermore, although both apigenin and naringenin alleviated hyperglycemia and hyperlipemia and insulin resistance, apigenin was more effective than naringenin at an equivalent dose [34]. It was also claimed that the inhibitory activity of apigenin against α-glucosidase was higher than that of naringenin [35]. Therefore, the metabolism of naringenin in vivo to produce apigenin may enhance pharmacological activity and synergize with naringenin to have stronger medicinal effects. Simultaneously, other compounds such as phloretin (**N1**) and *O*-desmethylangolensin (**N5**) with superior activity were also found in the metabolic process [36,37]. Furthermore, hippuric acid (**M35**) was detected in all biological samples in the metabolic pathway of naringenin. Naringenin undergoes ring fission in vivo to produce a series of organic acids, such as p-hydroxybenzoic acid and 3-(4-hydroxyphenyl) propionic acid, and is ultimately converted into hippuric acid and 4′-hydroxyhippuric acid [38].

### 3.2. Comparison of the Different Biological Treatment Methods

The main disadvantage of chromatographic methods is the need for sample preparation, which determines whether the sample can be used in the chromatograph in its original form. According to the accurate mass measurements, fragmentation patterns, diagnostic product ions, and literature reports, 31 metabolites were screened and identified in the plasma samples by using the UHPLC-HRMS method, i.e., eight in samples treated with the SPE solid-phase extraction method, 16 in samples treated with methanol precipitation, and 15 in samples treated with acetonitrile precipitation (Figure 6C). Among them, six identical metabolites were discovered in samples subjected to methanol precipitation and acetonitrile precipitation, and two identical metabolites were identified in samples subjected to SPE and methanol precipitation. In contrast, only one identical metabolite was detected in samples subjected to SPE and acetonitrile precipitation.

Although SPE can be used to extract and pre-concentrate a wide range of compound classes and is easy to automate to increase the reproducibility of extractions, this sample preparation method is complex and requires extensive organic solvent consumption. It also has some technical problems [39,40]. In this study, there were lower amounts of samples prepared by SPE than samples prepared by methanol precipitation and acetonitrile precipitation, meaning that SPE may not be suitable for in vivo metabolism studies of such substances. Furthermore, the samples obtained by methanol precipitation and acetonitrile precipitation were similar, so one of them could be selected during sample preparation. During the in vivo metabolic analysis of naringenin, the difference between SPE and the other two preparation methods was mainly reflected in glucuronidated metabolites and organic acid metabolites. More phenolic acid metabolites could be detected in samples prepared by SPE, while more glucuronidated metabolites could be detected in samples prepared by methanol precipitation and acetonitrile precipitation. Differences in metabolite quantity might be due to the different separation selectivity of organic solvents, leading to signal peaks with diverse intensity and quantity concerning naringenin metabolites after UHPLC-HRMS detection.

## 4. Materials and Methods

### 4.1. Chemicals and Reagents

Naringenin standard was purchased from Chengdu Must-Technology Co., Ltd. (Chengdu, China). Its purity was acceptable (≥98%) according to HPLC–UV analysis, and its structure was fully elucidated by comparing the spectral data (ESI-MS and ^1^H, ^13^C-NMR spectroscopy) with the literature. HPLC-grade acetonitrile, methanol, and formic acid (FA) were purchased from Thermo Fisher Scientific (Fair Lawn, NJ, USA). The deionized water used throughout the experiment was purchased from Watsons (Guangzhou, China). Oasis^®^ HLB C_18_-low solid-phase extraction cartridges (500 mg·6 mL^−1^, 60 µm, 149 Å) were purchased from Waters Corporation (Milford, MA, USA). Rat liver microsomes were obtained from NEWGAINBIO Co., Ltd. (Wuxi, China). Nicotinamide adenine dinucleotide phosphate (NADPH) and MgCl_2_ were purchased from Shanghai Macklin Biochemical Co., Ltd. (Shanghai, China). Six-well plates were obtained from Corning Incorporated-Life Science (Hangzhou, China).

### 4.2. In Vivo Experiment

#### 4.2.1. Animals and Drug Administration

Six male SD rats weighing 220 ± 20 g were obtained from Jinan Pengyue Experimental Animals Company (Jinan, China). The rats were randomly divided into two groups: the drug group (n = 3) for test plasma, urine, feces, and liver, and the control group (n = 3) for blank plasma, urine, feces, and liver. The rats in the drug group were given a dose of 255 mg·kg^−1^ body weight orally, administered for 3 d. A standard saline solution (2 mL) was administered to the rats in the control group every day. Other culture conditions were consistent with our previous report [27]. The animal facilities and protocols complied with the Guide for the Care and Use of Laboratory Animals (USA National Research Council, 1996).

#### 4.2.2. Sample Collection and Preparation

Plasma, urine, and feces samples were taken from rats and treated with SPE cartridges by the previous processing method [27]. After 24 h of the last drug administration, rat liver tissues were removed from dissected rats and quenched in liquid nitrogen, and then they were stored at −80 °C. During further processing, liver tissue (0.5 g) was ground with methanol and centrifuged to get the supernatant. The supernatant was added to the pretreated SPE cartridges [27], and then the same process described above was conducted. Furthermore, plasma samples were processed by two other methods so as to explore the impact of different processes of biological samples on metabolite detection: methanol and acetonitrile were added separately to reach a final concentration of 75%. These samples were precipitated for 30 min and then centrifuged at 3500 rpm for 15 min to obtain the solutions after treatment. Finally, all samples were dried under N_2_ at room temperature. The residue was then redissolved in 300 µL of methanol and centrifuged for 15 min (14,000 rpm, 4 °C). The obtained supernatant was used for further instrumental analysis.

### 4.3. Experiment In Vitro

The in vitro metabolism of naringenin was carried out in rat liver microsomes. A reaction mixture containing phosphate buffer (pH 7.4), MgCl_2_ (3 mM, final concentration), rat liver microsomes (1 mg·mL^−1^, final protein concentration), and naringenin (0.11 mg·mL^−1^, final concentration) was prepared and called the dosing solution. The drug-negative solution contained phosphate buffer (pH 7.4), MgCl_2_ (3 mM, final concentration), and rat liver microsomes (1 mg·mL^−1^, final protein concentration). In the 6-well plate, the dosing solution was added to the drug group, while the control group was given a drug-negative solution. After that, the drug group and the control group were pre-incubated in a water bath at 37 °C for 5 min. The reaction was started by adding NADPH (2.55 mg·mL^−1^, final concentration) dissolved in buffer. The incubation continued at 37 °C, and 100 μL of supernatant was taken out after 5, 10, 15, 30, 45, 60, 120, and 240 min. Subsequently, 200 μL cold acetonitrile was added to stop the reaction, followed by centrifugation at 3500 rpm for 15 min. Finally, the supernatant was dried under N_2_ at room temperature. The residue was then redissolved in 300 µL of methanol and centrifuged for 15 min (14,000 rpm, 4 °C). The obtained supernatant was used for further LC–MS analysis.

### 4.4. Instruments and Analytical Conditions

#### 4.4.1. UHPLC Parameters

Chromatographic separation was performed on a Vanquish column compartment equipped with a Vanquish autosampler (Thermo Electron, Bremen, Germany). Separation was performed on a Waters ACQUITY BEH C_18_ column (100 × 2.1 mm, 1.7 µm, 130 Å). A flow rate of 0.3 mL·min^−1^ was set to separate the drug metabolites. The column temperature was maintained at 30 °C, and the injection volume was 2 μL. The mobile phase was composed of acetonitrile (A) and water containing 0.1% formic acid (B). The gradient elution conditions were set as follows: 0–15.0 min, 95–60% B; 15.0–18.0 min, 60% B; 18.0–22.0 min, 60–30% B; 22.0–22.1 min, 30–95% B; 22.1–25.0 min, 95% B.

#### 4.4.2. High-Resolution ESI-MS (HRMS) Parameters

HRMS and MS/MS spectra were obtained using a Q-Orbitrap Exactive Plus mass spectrometer (Thermo Electron, Bremen, Germany) connected to the UHPLC instrument via a heated electrospray ionization (HESI) source. Mass spectrometric detection was performed in both positive and negative ion modes. The ion source parameters were as follows: nitrogen (purity ≥ 99.99%) served as the sheath gas and auxiliary gas at a flow rate of 45 and 10 (arbitrary units), respectively; a capillary temperature of 320 °C and spray voltage of 3800/3500 V (+/−) were used. HRMS was acquired at full scan in a mass range of *m*/*z* 80–1200 at a resolution of 140,000, while the resolution of dd-MS^2^ was 17,500.

### 4.5. Peak Selections and Data Processing

A Thermo Xcalibur 2.1 workstation was used for data acquisition and processing. To obtain as many ESI-MS/MS fragment ions of naringenin metabolites as possible, the peaks detected with an intensity higher than 40,000 for the positive ion mode and 10,000 for the negative ion mode were selected for identification. The chemical formulas attributed to the selected peaks were calculated using a formula predictor by setting the parameters as follows: C [5–30], H [5–60], O [2–20], S [0–2], N [0–3] and the ring double bond (RDB) equivalent value [3–20].

## 5. Conclusions

In this study, three methods of biological sample preparation were applied to analyze the in vivo and in vitro metabolism of naringenin. It is worth noting that more metabolites could be retained in samples treated with methanol precipitation and acetonitrile precipitation. Although SPE may not apply to all compounds, it could still be a crucial complement to sample preparation. Our study has demonstrated that compounds obtained by SPE were different from those obtained by other methods. In addition, a UHPLC-Q-Exactive Orbitrap mass spectrometer, which has high selectivity, specificity, and sensitivity [41], was used to investigate the in vitro and in vivo metabolic profiles of naringenin, and an analytical strategy of “ion induction and deduction” based on metabolic “net-hubs” was performed. Due to the difference in metabolic reactions, we divided the metabolic reactions into two categories: in the first, the mass spectral information of such metabolites matched well with that of naringenin, but this did not occur in the second. Therefore, we constructed multiple “net-hubs” and three DPI groups according to the different characteristics of these reactions. Then, the mass spectral information of samples was quickly compared with that of the DPI groups to rapidly screen possible naringenin metabolites.

Finally, 78 naringenin metabolites were identified from plasma, urine, feces, liver tissue, and liver microsome samples. The main biotransformation pathways observed were glucuronidation, sulfation, oxidation, methylation, and so on. Furthermore, we found that naringenin could undergo ring fission to generate naringenin chalcone, phloretin, and various aromatic acids such as 3-(4-hydroxyphenyl) propionic acid and hippuric acid. Metabolites such as apigenin can play a synergistic effect in the body, helping to exert stronger biological activity. Our results supply valuable data for a better understanding of naringenin and provide ideas for the analysis of the metabolites of other natural compounds.

## Figures and Tables

**Figure 1 molecules-27-07282-f001:**
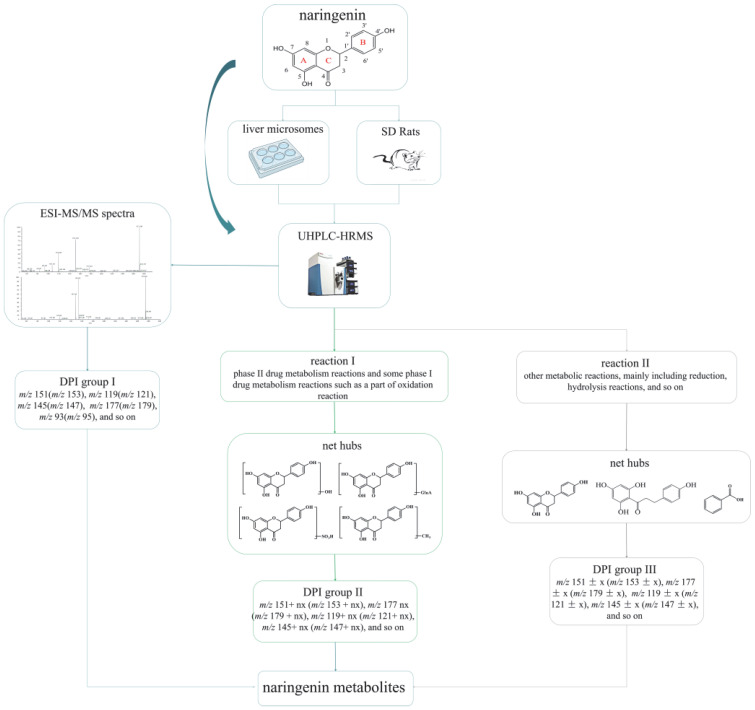
Flow diagram of naringenin metabolic analysis and the analytical strategy of “ion induction and deduction” based on metabolic “net-hubs”.

**Figure 2 molecules-27-07282-f002:**
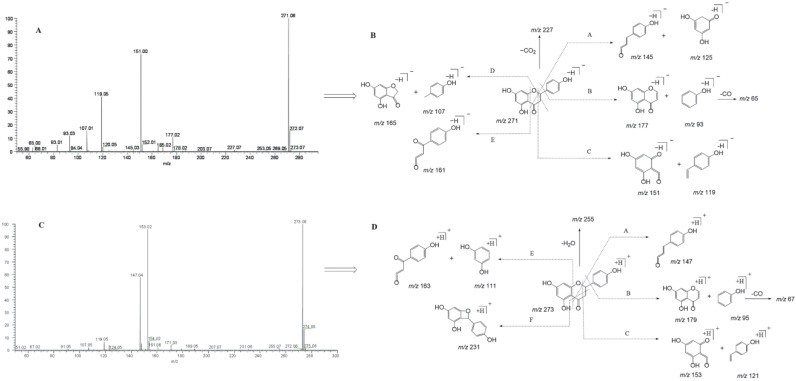
ESI-MS/MS spectral information and the cleavage pathways of naringenin. (**A**): ESI-MS/MS spectral information in negative mode; (**B**): cleavage pathways in negative mode; (**C**): ESI-MS/MS spectral information in positive mode; (**D**): cleavage pathways in positive mode.

**Figure 3 molecules-27-07282-f003:**
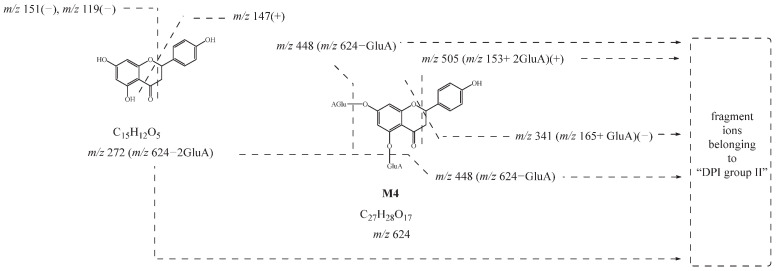
Possible cleavage pathway of **M4**.

**Figure 4 molecules-27-07282-f004:**
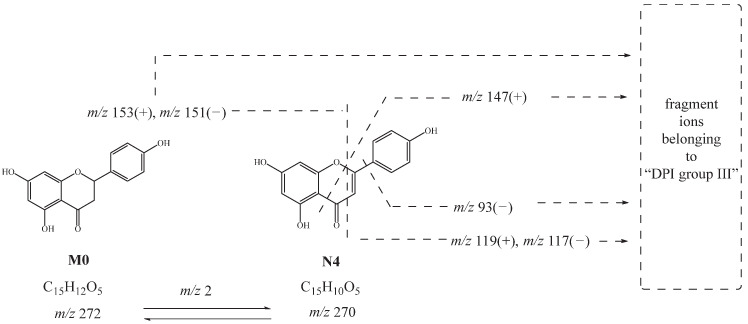
Possible cleavage pathway of **N4**.

**Figure 5 molecules-27-07282-f005:**
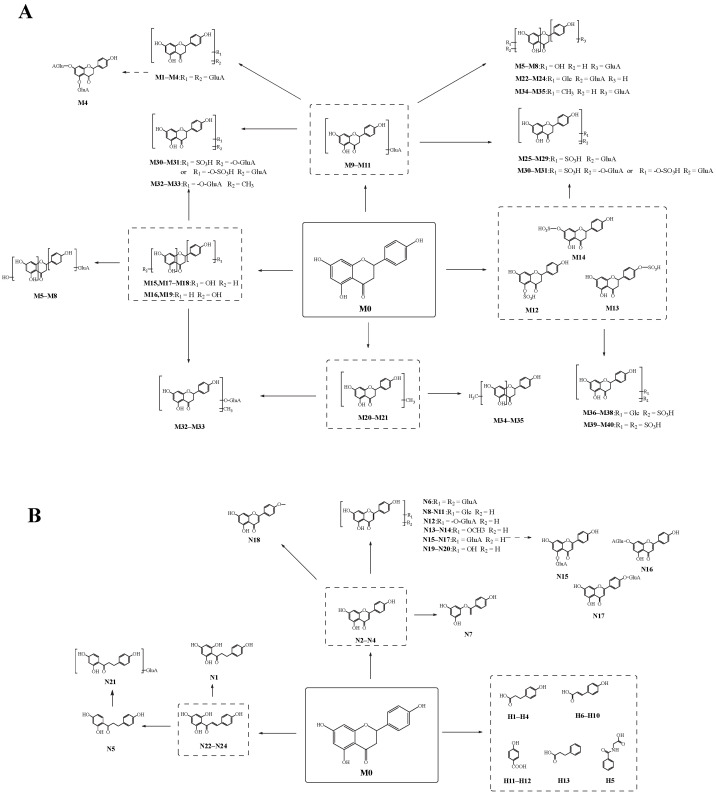
The possible metabolic pathways of naringenin. (**A**): the possible metabolic pathway of naringenin by “reaction I”; (**B**): the possible metabolic pathway of naringenin by “reaction II”.

**Figure 6 molecules-27-07282-f006:**
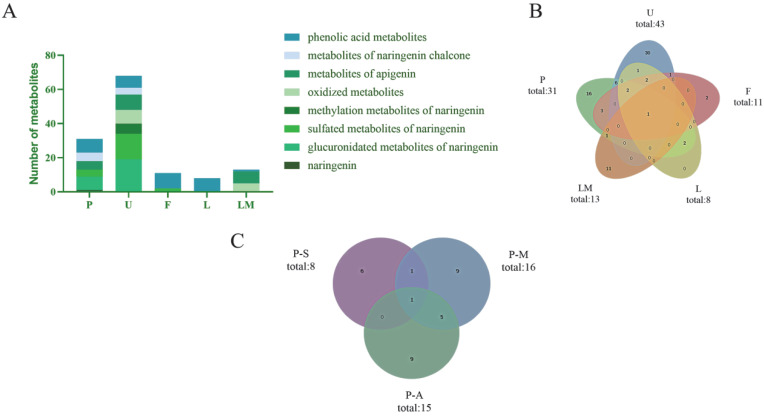
Distribution of naringenin metabolites in samples. (**A**): classification and distribution of metabolites in different metabolic samples (P: plasma; U: urine; F: feces; L: liver tissue; LM: liver microsomes); (**B**): quantity distribution of metabolites in different metabolic samples (P: plasma; U: urine; F: faces; L: liver tissue; LM: liver microsomes); (**C**): distribution of naringenin metabolites in samples with different preparation methods (P-S: plasma—SPF solid phase extraction method; P-M: plasma—methanol precipitation; P-A: plasma—acetonitrile precipitation).

**Table 1 molecules-27-07282-t001:** Metabolites of naringenin in rat plasma, urine, feces, liver, and liver microsomes.

Peak	tR (min)	Formula [M ± H]^+^	Ion Mode	Theoretical Mass (*m*/*z*)	Experimental Mass (*m*/*z*)	Error (ppm)	MS/MS Fragment Ions	P-S	P-M	P-A	U	F	L	LM
M0 *	13.33	C15H11O5	N	271.06120	271.06226	3.923	271.06 (100), 151.00 (69), 119.05 (40), 107.01 (16), 93.03 (11), 152.01 (3), 120.05 (3)	√						
	13.31	C15H13O5	P	273.07575	273.07599	0.879	153.02 (100), 273.08 (92), 147.04 (60), 154.02 (7), 148.05 (6), 119.05 (5), 123.04 (4)		√	√				√
Naringenin Metabolites by “Reaction I”														
M1	6.20	C27H27O17	N	623.12480	623.12726	3.029	447.09 (100), 113.02 (59), 271.06 (56), 623.13 (25), 151.00 (21), 175.02 (21), 119.05 (6)				√			
M2	6.8	C27H27O17	N	623.12480	623.12744	3.318	447.09 (100), 271.06 (53), 85.03 (26), 175.02 (20), 151.00 (20), 119.05 (6)		√	√	√			
M3	8.39	C27H27O17	N	623.12480	623.12756	3.511	271.06 (100), 151.00 (52), 447.09 (45), 119.05 (16), 175.02 (11), 85.03 (7), 107.01 (3)		√	√				
	8.35	C27H29O17	P	625.13935	625.14093	1.607	273.08 (100), 153.02 (43), 449.11 (34), 147.04 (28), 171.03 (10), 497.30 (3),		√					
M4	9.61	C27H27O17	N	623.12480	623.12689	4.196	623.13 (48), 271.06 (19), 151.00 (12), 175.02 (5), 119.05 (3), 447.09 (2), 459.09 (1), 341.73 (0.4)				√			
	9.56	C27H29O17	P	625.13935	625.13953	−0.633	273.08 (100), 153.02 (39), 171.03 (24), 147.04 (15), 431.10 (10), 413.09 (8), 505.08 (0.3), 497.10 (0.2)				√			
M5	8.23	C21H19O12	N	463.08770	463.08948	3.95	463.09 (100), 151.00 (95), 287.06 (90), 337.04 (45), 175.02 (27), 85.03 (22.42), 271.06 (13), 285.04 (7), 167.00 (6), 141.02 (4), 193.02 (3)				√			
M6	8.60	C21H19O12	N	463.08770	463.08926	3.475	287.06 (100), 337.04 (79), 463.09 (28), 167.00 (17), 125.02 (15), 175.02 (15), 119.05 (12), 271.06 (11), 235.15 (9), 193.03 (4), 141.02 (2)				√			
M7	9.57	C21H19O12	N	463.08770	463.08838	1.575	287.06 (100), 151.00 (75), 463.09 (56), 175.02 (26.32), 125.02 (16.63), 285.04 (13), 337.04 (6), 177.02 (5), 193.03 (3), 167.03 (1), 141.02 (1)				√			
M8	9.72	C21H19O12	N	463.08770	463.08896	1.445	287.06 (100), 463.09 (94), 285.04 (48.97), 151.00 (43), 175.02 (29), 337.04 (28), 119.05 (27), 167.00 (17), 193.01 (2), 141.02 (1)				√			
M9	7.97	C21H21O11	P	449.10725	449.10840	1.252	273.08 (100), 153.02 (52), 147.04 (21), 413.09 (3), 231.03 (2), 431.09 (1)		√		√			
M10	9.61	C21H19O11	N	447.09270	447.09467	3.099	271.06 (100), 151.00 (48), 175.02 (37), 447.09 (24), 271.03 (5)	√						
	9.60	C21H21O11	P	449.10725	449.10822	0.851	273.08 (100), 153.02 (31), 147.04 (22), 154.02 (3), 449.11 (2), 123.04 (2), 255.77 (1)	√	√	√	√			
M11	10.77	C21H21O11	P	449.10725	449.10822	0.851	273.08 (100), 153.02 (54), 147.04 (37), 449.11 (16), 123.04 (2), 179.03 (1), 255.07 (0.4), 231.07 (0.4)		√	√	√			
M12	10.75	C15H11O8S	N	351.01750	351.01852	3.012	271.06 (100), 351.02 (56), 151.00 (37), 207.14 (24), 119.05 (14), 177.02 (5), 79.96 (3), 107.01 (3), 225.15 (2)					√		
	10.78	C15H13O8S	P	353.03205	353.03278	0.610	273.08 (100), 153.02 (83), 353.03 (77), 147.04 (52), 171.03 (5), 121.10 (2), 335.22 (1), 175.11 (0.5), 203.11 (0.3), 227.00 (0.3)				√			
M13	11.10	C15H11O8S	N	351.01750	351.01907	3.016	271.06 (100), 351.02 (34), 151.00 (28), 177.02 (4), 119.05 (9), 93.03 (4), 107.01 (2)			√				
	11.14	C15H13O8S	P	353.03205	353.03268	0.327	273.08 (100), 353.03 (87), 153.02 (79), 147.04 (52), 335.22 (8), 121.10 (5), 109.10 (5), 317.21 (4), 175.11 (2), 203.14 (1), 191.11 (1), 231.06 (1)				√			
M14	12.04	C15H11O8S	N	351.01750	351.01901	2.845	271.06 (100), 151.00 (34), 351.02 (28), 119.05 (11), 107.01 (5), 93.03 (2), 227.07 (2)			√		√		
	12.06	C15H13O8S	P	353.03205	353.03275	0.525	273.08 (100), 153.02 (87), 353.03 (80), 147.04 (50), 109.10 (11), 121.10 (10), 95.09 (10), 171.03 (8), 335.22 (8), 203.14 (2), 175.11 (2)				√			
M15	9.58	C15H11O6	N	287.05556	287.05692	2.817	259.06 (100), 125.02 (99), 287.06 (83), 243.07 (19), 151.00 (17), 177.05 (10), 135.04 (4), 109.03 (3)							√
	9.55	C15H13O6	P	289.07011	289.07101	0.345	153.02 (100), 215.07 (59), 243.07 (56), 107.05 (53), 289.07 (42), 163.04 (16), 195.03 (10), 167.03 (10)							√
M16	9.99	C15H11O6	N	287.05556	287.05676	2.260	287.06 (100), 151.00 (34), 167.00 (26), 119.05 (18), 259.06 (14), 193.68 (3)							√
M17	10.38	C15H11O6	N	287.05556	287.05698	3.026	125.020 (100), 287.06 (17), 151.00 (12), 135.04 (10), 161.02 (5), 107.01 (2)							√
	10.38	C15H13O6	P	289.07011	289.07111	0.445	153.02 (100), 289.07 (15), 163.04 (1), 161.85 (1)							√
M18	11.52	C15H11O6	N	287.05556	287.05643	1.110	151.00 (100), 135.04 (68), 287.06 (20), 107.01 (10), 152.01 (8), 125.02 (4), 65.00 (4), 109.03 (1)							√
	11.51	C15H13O6	P	289.07011	289.07059	−0.075	289.07 (100), 153.02 (85), 163.04 (79), 179.03 (8), 111.01 (3), 139.04 (2), 151.04 (1), 229.05 (0.4), 137.06 (0.4)							√
M19	13.30	C15H11O6	N	287.05556	287.05594	−0.597	287.06 (100), 151.00 (99), 177.02 (27), 119.05 (24), 167.00 (21), 107.01 (13), 193.01 (6)							√
M20	11.04	C16H13O5	N	285.07630	285.07535	−3.327	241.14 (100), 285.13 (57), 179.14 (36), 242.15 (14), 160.04 (6), 97.06 (3), 205.12 (2), 177.13 (2), 83.05 (2), 151.11 (2), 107.05 (0.4)				√			
M21	14.85	C16H13O5	N	285.07630	285.07693	2.216	160.04 (32), 179.03 (29), 285.08 (28), 253.15 (27), 241.08 (16), 205.03 (7), 83.05 (4)				√			
M22	6.03	C27H29O16	N	609.14548	609.14838	3.730	343.08 (100), 609.15 (85), 313.07 (61), 519.12 (53)?, 271.06 (16), 591.14 (14), 489.10 (13), 573.13 (11), 119.05 (10), 555.11 (8), 151.00 (8), 433.12 (8), 448.10 (3)			√				
	6.04	C27H31O16	P	611.16004	611.16101	0.571	436.14 (83), 315.09 (37), 417.12 (18), 285.08 (15), 381.10 (14), 147.04 (10), 399.11 (10), 273.08 (6), 473.11(4)			√				
M23	6.61	C27H29O16	N	609.14548	609.14838	3.730	609.15 (100), 313.07 (73), 343.08 (44), 489.10 (20), 271.06 (13), 433.12 (12), 151.00 (7), 119.05 (7)			√				
	6.61	C27H31O16	P	611.16004	611.16119	0.865	315.09 (36), 417.12 (20), 285.08 (17), 436.14 (10), 399.11 (5)			√				
M24	7.66	C27H29O16	N	609.14548	609.14819	3.418	313.07 (100), 609.15 (97), 489.10 (38), 343.08 (37), 271.06 (12), 433.11 (7), 151.00 (5), 119.05 (5)			√				
	7.67	C27H31O16	P	611.16004	611.16138	1.176	315.09 (31), 285.08 (18), 417.12 (14), 399.11 (11), 273.08 (8), 147.04 (6)			√				
M25	6.03	C21H19O14S	N	527.04958	527.05145	2.563	271.06 (100), 527.05 (51), 151.00 (35), 447.09 (23), 351.02 (11), 119.05 (7), 175.02 (5), 107.01 (2), 177.02 (2), 227.07 (2)				√			
M26	6.38	C21H19O14S	N	527.04958	527.05109	1.880	271.06 (100), 527.05 (56), 151.00 (34), 447.09 (25), 351.02 (9), 119.05 (7), 175.02 (5), 107.01 (2)				√			
M27	7.93	C21H19O14S	N	527.04958	527.05096	1.633	271.06 (100), 351.02 (73), 527.05 (62), 447.09 (48), 151.00 (26), 175.02 (17), 119.05 (5.88), 177.02 (4), 93.03 (3)				√			
M28	8.42	C21H19O14S	N	527.04958	527.05157	2.791	271.06 (100), 351.02 (63), 527.05 (54), 447.09 (40), 151.00 (25), 175.02 (15), 119.05 (7), 177.02 (3), 107.01 (2)				√			
M29	9.50	C21H19O14S	N	527.04958	527.05133	2.336	271.06 (60), 527.05 (57), 351.02 (175.02 (28 ), 447.09 (21), 151.00 (14), 119.05 (4), 75.01 (3), 177.02 (2), 93.03 (1)				√			
M30	7.76	C21H19O15S	N	543.04448	543.04633	2.424	367.01 (100), 151.00 (87), 287.06 (76), 543.05 (46), 463.09 (38), 175.02 (12), 107.01 (2), 271.06 (1)				√			
M31	8.25	C21H19O15S	N	543.04451	543.04584	1.522	367.01 (81), 151.00 (77), 287.06 (73), 351.02 (44), 463.20 (44), 271.06 (37), 175.02 (28), 368.02 (12), 79.96 (12)				√			
M32	9.94	C22H21O12	N	477.10328	477.10501	2.433	301.07 (100), 271.06 (70), 151.00 (46), 477.10 (40), 175.02 (27), 399.05 (6), 107.01 (6)177.02 (4), 286.05 (4)				√			
	9.96	C22H23O12	P	479.11784	479.11862	−1.835	303.09 (100), 177.05 (34), 153.02 (31), 179.03 (5), 479.11 (3), 147.04 (3)				√			
M33	10.34	C22H21O12	N	477.10328	477.10489	2.181	271.06 (85), 301.07 (80), 477.10 (55), 175.02 (22), 151.00 (12), 286.05 (8), 399.05 (7)				√			
	10.33	C22H23O12	P	479.11784	479.11880	−1.460	303.09 (100), 177.05 (55), 153.02 (36), 179.03 (8), 479.12 (3), 154.02 (2), 121.04 (1)				√			
M34	11.83	C22H21O11	N	461.10838	461.11020	2.744	271.06 (100), 461.11 (33), 151.00 (31), 119.05 (13), 337.04 (10), 285.08 (8), 175.00 (7), 401.09 (5), 93.03 (5), 177.02 (4)				√			
M35	13.28	C22H21O11	N	461.10838	461.11014	2.614	285.08 (100), 175.02 (21), 461.11 (13), 337.04 (4), 243.07 (4), 271.06 (4), 151.00 (3), 301.07 (1)				√			
M36	6.06	C21H21O13S	N	513.07021	513.07202	2.311	271.06 (100), 513.07 (54), 433.11 (35), 151.00 (30), 177.02 (3), 93.03 (2), 241.00 (2), 227.07 (1)				√			
M37	6.17	C21H21O13S	N	513.07021	513.07233	2.915	271.06 (100), 513.07 (48), 433.11 (32), 151.00 (31), 119.05 (7), 177.02 (3), 241.00 (2), 107.01 (2), 93.03 (2)				√			
M38	6.50	C21H21O13S	N	513.07021	513.07178	1.843	271.06 (100), 513.07 (51), 433.11 (32), 151.00 (31), 177.02 (3), 241.00 (3), 107.01 (2)				√			
M39	13.56	C15H11O11S2	N	430.97431	430.97592	2.539	271.06 (100), 351.02 (86), 430.20 (15), 151.00 (14), 119.05 (4), 177.02 (2), 243.82)(2)		√		√			
M40	14.23	C15H11O11S2	N	430.97431	430.97586	2.400	271.06 (100), 351.02 (86), 151.00 (16), 349.00 (4), 177.02 (3), 93.03 (2)		√		√			
Naringenin Metabolites by “Reaction II”														
N1	1.23	C15H13O5	N	273.07685	273.07498	−4.828	179.84 (100), 227.07 (36), 273.83 (33), 125.87 (13)			√				
N2	8.93	C15H11O5	P	271.06005	271.06015	0.185	271.06 (100), 153.02 (7), 215.07 (5), 243.07 (4), 149.02 (2), 253.05 (2), 147.04 (1), 145.03 (1), 119.05 (1)				√			
N3	12.95	C15H11O5	P	271.06005	271.06009	−0.037	273.08 (100), 153.02 (88), 147.04 (51), 177.05 (10), 235.13 (10), 227.11 (10), 93.07 (8), 119.05 (7), 107.05 (6), 171.03 (5)				√			
N4	13.35	C15H9O5	N	269.04550	269.04657	3.804	151.00 (100), 107.01 (28), 93.03 (26), 65.00 (14), 125.02 (2), 269.05 (2), 117.03 (0.4)							√
	13.33	C15H11O5	P	271.06005	271.06070	2.214	153.02 (100), 147.04 (60), 271.06 (12), 119.05 (7), 171.03 (4), 107.05 (2), 151.04 (1)							√
N5	11.55	C15H15O4	P	259.09709	259.09677	1.098	149.06 (100), 121.07 (69), 165.05 (21), 137.06 (16), 259.10 (14), 213.09 (9), 122.07 (6), 241.09 (5), 139.06 (4)				√			
	11.53	C15H13O4	N	257.08254	257.08179	−0.553	221.84 (53), 257.08 (32), 135.04 (31), 239.13 (12), 211.13 (6), 151.11 (4)				√			
N6	5.56	C27H27O17	P	623.12370	623.12445	0.280	271.06 (100), 447.09 (31), 448.09 (6), 299.07 (1), 284.06 (0.5), 328.06 (0.4), 429.08 (0.3)				√			
N7	5.88	C14H11O4	N	243.06575	243.06673	1.843	243.07 (100), 124.02 (9), 149.02 (7), 225.06 (3), 93.03 (2), 122.89 (2)							√
N8	7.09	C21H21O10	P	433.11230	433.11313	−2.055	257.08 (100), 123.04 (52), 271.06 (42), 433.28 (38), 163.04 (30), 223.08 (24), 95.05 (6), 124.05 (4), 136.05 (2)				√			
N9	7.24	C21H21O10	P	433.11230	433.11298	−2.401	257.08 (100), 123.04 (54), 163.04 (31), 135.04 (6), 95.05 (6), 137.02 (5), 271.06 (3), 433.11 (2)				√			
N10	8.06	C21H21O10	P	433.11230	433.11334	−1.570	257.08 (100), 123.04 (46), 163.04 (30), 95.05 (6), 135.04 (6), 137.02 (6), 253.14 (3), 209.15(2), 153.02 (1), 271.06 (1)				√			
N11	13.04	C21H21O10	P	433.11230	433.11340	−1.432	257.08 (100), 153.02 (21), 433.11 (2), 215.07 (1), 125.10 (1), 95.09 (0.5), 271.06 (0.3)				√			
N12	7.38	C21H17O12	N	461.07205	461.07431	3.819	285.04 (100), 461.07 (22), 177.02 (13), 151.00 (6), 175.02 (5), 107.01 (2), 241.05 (1)		√	√				
	7.38	C21H19O12	P	463.08660	463.08768	1.247	287.06 (100), 269.04 (7), 463.09 (6), 153.02 (3), 241.05 (2), 121.03 (1), 231.06 (1), 259.06 (1)		√					
N13	7.88	C16H11O6	N	299.05555	299.05670	1.968	163.00 (100), 135.04 (88), 299.06 (19), 281.05 (2), 123.95 (1), 161.84 (1), 93.03 (1)							√
N14	12.63	C16H11O6	N	299.05555	299.05664	1.768	163.00 (100), 135.04 (99), 299.06 (23), 119.05 (11), 164.01 (9), 281.05 (1), 93.03 (1)							√
N15	8.32	C21H17O11	N	445.07705	445.07907	3.225	269.05 (100), 175.02 (16), 445.08 (11), 151.00 (1), 327.03 (1), 161.51 (1), 119.03 (1)		√	√				
	8.31	C21H19O11	P	447.09160	447.09280	1.369	271.06 (100), 447.09 (10), 153.02 (5), 147.04 (2), 215.07 (1), 243.06 (1), 161.73 (0.3)		√					
N16	8.96	C21H17O11	N	445.07705	445.07910	3.293	269.05 (100), 445.08 (25), 175.02 (5), 225.06 (4), 151.00 (4), 311.06 (2)		√					
	8.92	C21H19O11	P	447.09160	447.09216	−0.062	271.06 (100), 447.22 (8), 95.09 (1), 225.11 (1), 215.07 (1), 176.11 (0.5), 253.05 (0.3), 313.07 (0.2), 329.07 (0.1)				√			
N17	9.47	C21H17O11	N	445.07705	445.07919	3.495	269.05 (100), 175.02 (15), 445.08 (11), 151.00 (2), 338.82 (1), 93.53 (1)		√					
	9.46	C21H19O11	P	447.09160	447.09274	1.235	271.06 (100), 447.09 (16), 153.02 (2), 147.04 (1)		√		√			
N18	11.42	C16H13O5	P	285.07520	285.07568	−0.246	285.08 (100), 270.05 (14), 133.09 (6), 225.05 (2), 177.11 (2)			√				√
N19	11.53	C15H9O6	N	285.04100	285.04153	3.749	151.00 (100), 135.04 (69), 107.01 (11), 125.02 (4), 285.04 (4), 65.00 (3), 257.05 (1)							√
N20	11.74	C15H9O6	N	285.04100	285.04105	2.065	285.04 (100), 151.00 (11), 135.04 (5), 107.01 (2), 241.05 (1)							√
	11.72	C15H11O6	P	287.05555	287.05530	0.285	287.05 (100), 153.02 (29), 163.04 (26), 135.04 (3), 179.03 (2), 123.04 (1), 271.06 (1), 151.04 (0.4)							√
N21	11.52	C21H23O10	P	435.12918	435.12906	1.119	149.06 (100), 259.10 (82), 121.07 (41), 213.09 (6), 241.09 (4), 271.19 (1), 435.12 (1), 95.09 (1)		√		√			
N22	9.59	C15H13O5	P	273.07580	273.07596	0.210	153.02 (100), 273.08 (96), 147.04 (50), 154.02 (8), 119.05 (6), 179.03 (2)		√					
N23	10.79	C15H13O5	P	273.07580	273.07596	0.210	153.02 (100), 273.08 (89), 147.04 (51), 154.02 (8), 119.05 (7), 237.16 (3)		√					
N24	8.33	C15H13O5	P	273.07580	273.07605	1.099	153.02 (100), 147.04 (57), 154.02 (7), 119.05 (6), 148.05 (6), 179.03 (1), 109.10 (1), 95.09 (1)		√					
Phenolic Acid Metabolites														
H1	7.12	C9H9O3	N	165.05572	165.05528	−2.650	147.04 (100), 165.06 (38), 121.03 (36), 119.05 (39), 137.03 (6), 103.92 (6)			√	√	√	√	
	7.08	C9H11O3	P	167.07027	167.07076	2.929	139.98 (69), 167.98 (35), 107.09 (27), 109.06 (18), 93.07 (18), 121.07 (15.01), 149.10 (13), 95.05 (12)	√						
H2	7.46	C9H9O3	N	165.05572	165.05525	−2.832	121.06 (100), 119.05 (51), 121.03 (47), 165.05 (24), 147.04 (5), 137.03 (5), 103.92 (4)			√	√	√	√	
	7.42	C9H11O3	P	167.07027	167.07057	1.791	120.08 (100), 139.98 (69), 167.98 (41), 107.09 (35), 102.97 (25), 93.07 (22), 109.07 (16.85), 121.07 (16)	√						
H3	8.46	C9H9O3	N	165.05572	165.05527	−2.711	121.06 (100), 165.05 (22), 119.05 (10), 147.04 (10), 93.03 (0.4)				√	√	√	
	8.49	C9H11O3	P	167.07027	167.07045	1.073	120.08 (100), 139.98 (60), 167.98 (34), 107.09 (32), 93.07 (20), 167.06 (18), 109.07 (18)	√						
H4	14.04	C9H9O3	N	165.05572	165.05504	−4.105	121.06 (100), 119.05 (63), 165.05 (30), 147.04 (13), 93.03 (4)				√	√	√	
	14.03	C9H11O3	P	167.07027	167.07040	0.774	120.08 (100), 139.98 (60), 167.98 (34), 107.09 (32), 93.07 (20), 109.07 (18), 167.06 (17)	√						
H5	4.85	C9H8NO3	N	178.05097	178.05054	−2.395	134.06 (100), 178.05 (65), 160.04 (1), 121.03 (1), 102.03 (1), 77.04 (0.3)	√	√	√	√	√	√	√
	4.84	C9H10NO3	P	180.06552	180.06580	−4.534	105.03 (100), 180.07 (1), 77.04 (1), 162.06 (0.1), 136.02 (0.1), 79.06 (0.1)	√	√	√	√	√	√	√
H6	1.10	C9H9O3	P	165.05462	165.05482	1.208	123.04 (100), 119.05 (38), 95.05 (30), 147.04 (18), 103.05 (3), 121.07 (2)	√	√			√		
H7	5.73	C9H9O3	P	165.05462	165.05495	1.995	165.05 (100), 137.06 (13), 109.07 (9), 95.05 (6)	√				√		
H8	9.06	C9H9O3	P	165.05462	165.05487	1.511	165.05 (100), 120.08 (14), 137.06 (13), 109.07 (8), 95.05 (6)	√						
H9	6.77	C9H7O3	N	163.04007	163.03955	−3.174	119.05 (100), 163.04 (13), 118.03 (1), 93.03 (1)				√	√		
H10	8.15	C9H7O3	N	163.04007	163.03954	−3.235	119.05 (100), 163.04 (32), 115.92 (1), 93.03 (1), 135.04 (0.5)				√		√	
H11	2.97	C7H5O3	N	137.02442	137.02374	−4.943	93.03 (100), 137.02 (57), 109.03 (46), 65.01 (9), 119.02 (5)	√					√	
H12	9.38	C7H5O3	N	137.02442	137.02373	−1.014	93.03 (100), 137.02 (36), 109.03 (24), 119.02 (5)	√					√	
H13	11.78	C9H9O2	N	149.06080	149.06003	−1.506	149.06 (100), 121.03 (25), 91.03 (2), 119.05 (2), 105.04 (2)					√		

Note: t_R_: retention time; P: positive ion mode; N: negative ion mode; *: standard substance; √: metabolite was detected; P-S: plasma—SPF solid phase extraction method; P-M: plasma—methanol precipitation; P-A: plasma—acetonitrile precipitation; U: urine; F: feces; L: liver tissue; LM: liver microsome.

## Data Availability

Most of the data used during the preparation of the manuscript are included in the Results and Discussion sections. However, for any additional details of the procedures and the original raw files, please contact the corresponding authors.

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
