# Peer review of "The Analytical Strategy of “Ion Induction and Deduction Based on Net-Hubs” for the Comprehensive Characterization of Naringenin Metabolites In Vivo and In Vitro Using a UHPLC-Q-Exactive Orbitrap Mass Spectrometer"

_molecules, 2022, doi:10.3390/molecules27217282_

Round 1

Reviewer 1 Report

Manuscript ID: molecules-1987778

The authors described the characterization of different metabolites of Naringenin flavones subclass.

The paper is very interesting and the use of Orbitrap exact mass to classify the different classes of metabolites is very accurate.

The only suggestion is to strongly reduce the descriptive part of the results. I can imagine the huge work to classify all the fragment from different biological systems, but form the point of view of the readers this part is quite boring. Maybe the authors could enhance the description using the figure of the metabolites and enlarge the structure to permit the readers to better understand all the biochemical modification. The description of the single ion and the associate mass doesn’t add information.

One minor concern:

Pag 4 chapter 2.4.1;  better describe the mobile phase for UPLC, is not clear which solvent is A and B.

Reviewer 2 Report

this manuscript proposed the use of ion induction and deduction to systematically screen and identify naringenin metabolites in vivo and in vitro with UHPLC-Q_exactive orbitrap mass spectrometer. As a result, 78 naringenin metabolites were detected and identified from different samples matrices including plasma, urine, feces, liver tissue, and liver microsomes. Overall, I find this article well-written and comprehensively structured.From my perspective, it is suitable for publication after addressing some questions I’ve got in the following:

1. At page 2, Definitions and backgrounds for “net hubs” should be presented in the introduction to help reader better understand the strategy. For instance, what is the difference between the proposed net hubs and other reported network strategies? What is the advantages?

2. Page 4, for 2.4.1. UHPLC parameters , please clarify the pore size for the column?Also,  should the particle size be 1.7um? 
